# The Effect of Climate Change on the Water Supply and Hydraulic Conditions in the Upper Pejibaye River Basin, Cartago, Costa Rica

Fernando Watson-Hernández [1,*], Isabel Guzmán-Arias [1], Laura Chavarría-Pizarro [1] and Francisco Quesada-Alvarado [2,3]

1    Instituto Tecnológico de Costa Rica, Cartago 30101, Costa Rica; iguzman@itcr.ac.cr (I.G.-A.); laura.chavarria@itcr.ac.cr (L.C.-P.)
2    Programa Regional de Posgrado en Biología, Sistema de Estudio de Posgrado, Escuela de Biología, Universidad de Costa Rica (UCR), San José 11501-2060, Costa Rica; francisco.quesada.alvarado@una.ac.cr
3    Instituto Regional de Estudios en Sustancias Tóxicas (IRET), Universidad Nacional (UNA), Heredia 40101, Costa Rica
*    Correspondence: fwatson@itcr.ac.cr; Tel.: +50-68-474-6643

**Abstract:** The consequences of climate change have challenged researchers to generate models and projections to understand climate behavior under different scenarios. In Costa Rica, as in other countries, climate-change (CC) models and projections are essential to make decisions about the management of natural resources, mainly water. To understand climate change's impact on hydraulic parameters such as velocity, depth, and river surface area, we studied the Pejibaye river basin, located in Jiménez in Cartago, Costa Rica. This watershed is characterized by having more than 90% of its surface area covered by forest. We used the precipitation and temperature data from meteorological stations (2000 to 2009) and climate-change scenarios (2000–2099) to predict the response of the basin in different periods. First, we calibrated (*NSE* = 0.77) and validated (*NSE* = 0.81) the HBV hydrological model using ten years of daily data from 2000 to 2009. The climate-change data (2000–2099) were incorporated into the calibrated HBV model. This allowed us to determine the impact of CC on the basin water regime for the periods 2040–2059 (CCS1) and 2080–2099 (CCS2). The IBER mathematical model was used to determine the changes in the hydraulic variables of the river flow. For the CCS1, we determined a 10.9% decrease in mean velocity and a 0.1-meter decrease in depth, while for CCS2, the effect will be an 11.3% reduction in mean velocity and a 0.14-meter decrease in depth. The largest decreases in river surface area per kilometer will occur in May (1710 m$^2$) for CCS1 and April (2250 m$^2$) for CCS2.

**Keywords:** Pejibaye river; climate change; hydrological variables

## 1. Introduction

Climate change has effects on the natural processes that take place on the planet and on the activities that humans develop on it. The characteristics of climate change depend to a substantial extent on the level of global warming, and the effects of this phenomenon on the climate fluctuate according to geographic location. The hydrological cycle is vulnerable to changes in climatological variables because it is directly affected by them. The hydrological cycle is intensifying, increasing precipitation and associated floods, as well as severe droughts in other regions [1].

In Costa Rica, the National Meteorological Institute [2], in 2022, published climate-change scenarios for each region of the country. These climate scenarios use climate data from 1961–1990 as reference points and project them to the future period of 2071–2100. In the Caribbean region, an increase in precipitation was expected on the coast, and accentuated in the central and southern sectors; in the mountainous zone, it was expected to remain

stable. In the area near the Central Eastern Valley, the trend was toward a reduction in the annual precipitation. The projected maximum and minimum temperatures progressively increased throughout the region. The maximum increase was between 2 and 4 °C, while the minimum increase was between 2 and more than 3 °C [3].

Hidalgo and Alfaro [4] evaluated 107 climate model runs, according to their capacity to reproduce the basic characteristics of the monthly precipitation and temperature variables (1979–1999), and determined that the best-rated model run was cesm1_cam5 (Euro-Mediterranean Center for Climate Change model). Following the methodology described by Hidalgo, Alfaro, and Quesada-Montano [5], Hidalgo and Alfaro [4] generated monthly climate projections for Costa Rica for the period 1979–2099, at a scale of 5.5 km × 5.5 km. The climate projection for the Pejibaye River basin, according to Hidalgo et al. [5], shows that for the last two decades of the 21st century, the annual mean temperature will increase by 2.5 °C and precipitation will decrease by 3.01%.

Simulations of climate change's impact on watersheds make it possible to establish lines of action for better management of forest, agricultural, industrial, and human resources, taking into consideration the current habitat in various watercourses [6]. This is possible regardless of whether the basins are located in more conserved regions with little human intervention, such as the Pejibaye River, or in basins with greater intervention, such as those in the central sector of the country. For example, the behavior of the hydrological cycle and the cumulative impact of human activities, such as sediments, pollution, and nutrients, along with the flow system, are relevant aspects with strong implications for the distribution of water for different uses. For these reasons, our objective was to determine the flow regime of the Pejibaye River by considering the hydrological and hydraulic components in the construction of climate-change scenarios. We considered climate-change variables to determine, among other aspects, the environmental flow required, following a hydrobiological methodology, to help understand the conditions that must be guaranteed to maintain the ecosystemic benefits of this river [7].

## 2. Materials and Methods

### 2.1. Study Area

The Pejibaye river basin is located in Jiménez, which belongs to Cartago province, and to the Upper Reventazón River Basin system located on the Caribbean slope of Costa Rica. Its approximate area is 250 km$^2$, and its altitude ranges from 570 to 2680 m.a.s.l., presenting areas with slopes greater than 60% (Figure 1).

In total, 80% of this basin is covered by dense forest because a large part of the area belongs to Tapantí National Park. The predominant land uses include sugar cane, pasture, coffee, and weeds. It is also known for having an average annual rainfall between 2800 and 8000 mm; in the central-western part of the basin, rainfall can reach 9000 mm/year, which makes it one of the rainiest areas of the country and a place with great hydroelectric potential, with an average annual flow of 34 m$^3$/s. Additionally, it is one of the areas that contribute most to the ICE's Angostura Hydroelectric Project [8].

### 2.2. Hydrologic Model

The model was supplied with daily data on temperature (°C), precipitation (mm), flow (m$^3$/s), monthly averages of temperature and evapotranspiration, and a matrix of land-use area fraction by elevation zone. The calibration was developed by using the Monte Carlo method (500,000 runs) through a uniform distribution within the given ranges for each parameter [9,10] (Table 1), selecting as the target function the Nash Sutcliffe model efficiency coefficient (*NSE*). The suggested ranges of the calibration parameters are presented in the following table:

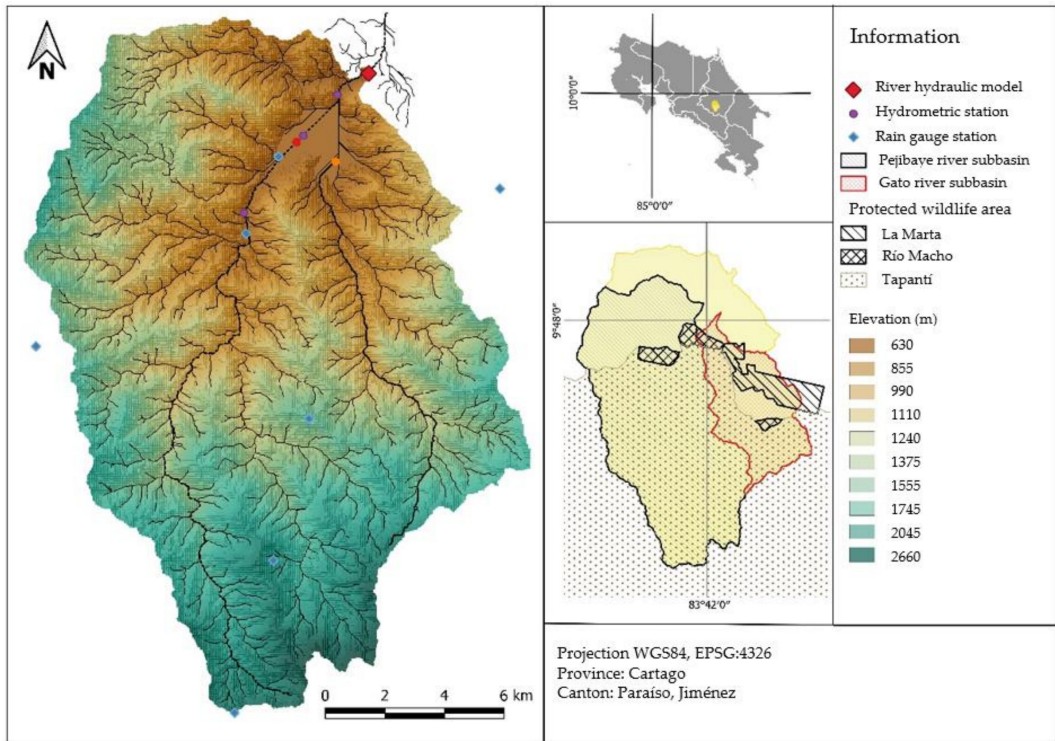

**Figure 1.** Location of the study and sampling points, protected wildlife areas, and subdivision of watersheds on the Pejibaye River basin, Cartago, Costa Rica.

**Table 1.** Range of parameters used for the calibration of the hydrological model.

| Parameter | Explanation | Minimum | Maximum | Units |
|-----------|-------------|---------|---------|-------|
| FC | Maximum soil humidity (ground box storage) | 50 | 500 | mm |
| LP | Evaporation reduction range (SM/FC) | 0.3 | 1 | - |
| BETA | Shape coefficient | 1 | 6 | - |
| CET | Correction factor for potential evaporation | 0 | 0,3 | 1/°C |
| | Response routine | | | |
| K1 | Recession coefficient (top box) | 0.01 | 0,4 | 1/d |
| K2 | Recession coefficient (bottom box) | 0.001 | 0,15 | 1/d |
| PERC | Maximum flow from upper to lower case | 0 | 3 | mm/d |
| MAXBAS | Routing, length of the ponderation function | 1 | 7 | d |

The program returned a file with the 100 best runs, evaluated from the comparison of observed flows (Qobs) and simulated flows (Qsim) using the *NSE* and containing the values of the calibration parameters that allowed the best adjustment. The procedures required to generate and calibrate the hydrological model of the basin are shown in Figure 2.

Having calibrated the model, it was validated using data from the last year of the flow series. In the evaluation of the models, we used the *NSE* coefficient, which measured how much of the variability of the observations was explained by the simulation, as well as the *PBIAS* coefficient, or bias percentage, which measured the probability that the average of the simulated values would be higher or lower than the observed values. The optimal bias value was zero, positive values meant underestimation by the model, and negative values indicated overestimation. The rating ranges used were taken from Moriasi et al. [11], and are shown in Table 2.

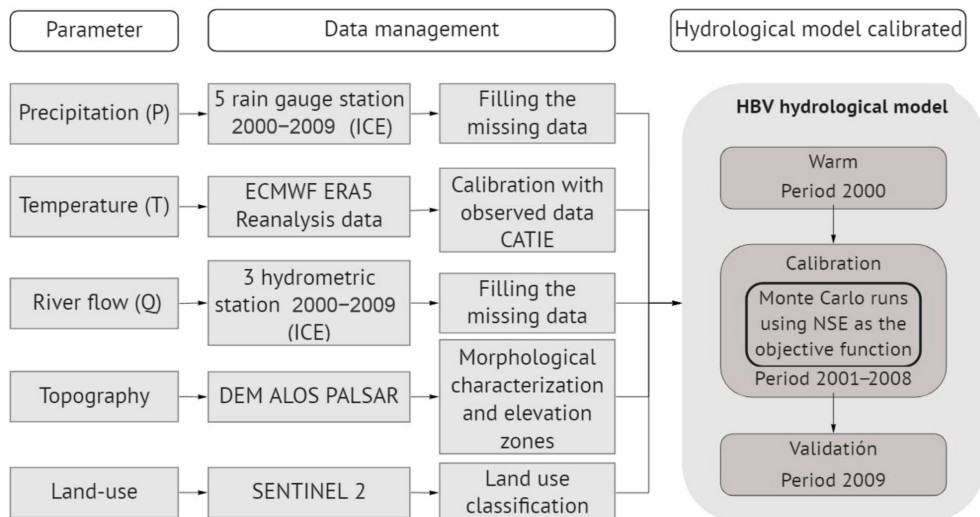

**Figure 2.** Diagram of the hydrologic model generation and calibration process.

$$NSE = 1 - \frac{\sum_{i=1}^{n}\left(y_i^{obs} - y_i^{sim}\right)^2}{\sum_{i=1}^{n}\left(y_i^{obs} - \overline{y}^{obs}\right)^2} \qquad (1)$$

$$PBIAS = \frac{\sum_{i=1}^{n}\left(y_i^{obs} - y_i^{sim}\right)\cdot 100}{\sum_{i=1}^{n}\left(y_i^{obs}\right)} \qquad (2)$$

$$R^2 = \left(\frac{\sum\left[\left(y_i^{obs} - \overline{y}^{obs}\right)\left(y_i^{sim} - \overline{y}^{sim}\right)\right]}{\sqrt{\sum\left(y_i^{obs} - \overline{y}^{obs}\right)^2 * \sum\left(y_i^{sim} - \overline{y}^{sim}\right)^2}}\right)^2 \qquad (3)$$

**Table 2.** Hydrological model evaluation ranges according to *NSE* and *PBIAS* statistics by Moriasi et al. [11].

| Evaluation | NSE | PBIAS (%) |
|---|---|---|
| Very good | 0.75 < NSE < 1.00 | PBIAS < ±10 |
| Good | 0.65 < NSE < 0.75 | ±10 < PBIAS < ±15 |
| Satisfactory | 0.50 < NSE < 0.65 | ±15 < PBIAS < ±25 |
| Unsatisfactory | NSE < 0.50 | PBIAS > ±25 |

*2.3. Climate-Change Scenario*

The climate change scenario (CCS) was obtained from the 2000–2099 monthly scale precipitation and temperature data generated by Hidalgo and Alfaro [4]. This CCS was corrected by adjusting the mean and standard deviation with data observed in the period 2000–2009. The data series obtained was called corrected CCS.

Subsequently, the CCS data corrected from monthly scale were disaggregated to daily values, using the Stochastic Weather Generator program, which is known by its acronym, WeaGETS [12]. Furthermore, a MATLAB code was created to pair and order the generated data with the CCS data, obtaining precipitation (P) and temperature (T) on a daily scale.

The WeaGETS model runs on the MATLAB platform and it required, as input information, the observed daily P and T data for the period 2000–2009 (10 years). The stochastic methods were established, and 1000 years of daily T and P data were generated. This information does not have an annual order, only a monthly order, and is based on the statistics of the observed data.

With a MATLAB code, which required as inputs the 1000 years of information (daily P and T) and the monthly corrected CCS, it was possible to compile a series of daily data whose statistics agreed with the CCS corrected on a monthly scale. The model works by taking month-by-month CCS data and searching among the stochastic data for the month with the highest similarity. For example, in the first cycle, it takes January of 2000 from the CCS for P and T (monthly cumulative and monthly daily average, respectively) and compares them with 1000 stochastically generated Januarys, choosing the one with the smallest difference. This action is repeated until the corrected CCS data series is completed.

The spatial scale of the CCS data is 5.5 km. This information was entered into the calibrated hydrological model to generate the 2000–2099 flow series and to determine the response of the basin. The flow data were divided into 20-year periods, 2000–2019, 2040–2059, 2080–2099, and water regimes were then generated for each period.

### 2.4. Hydraulic Model

The simulation of hydraulic conditions was developed using the IBER two-dimensional numerical model [13], requiring data such as topography derived from a Digital Elevation Model (DEM), boundary conditions, and roughness coefficient of the section under study. The DEM was generated from a topographic survey carried out with a total station, taking cross-sections separated every 5 m. Contour lines were constructed with the x, y, and z data points using the Qgis program and, after an editing process, were utilized to generate the DEM.

The contour conditions upstream were established as a step input, which is not a dependency of time, but the fulfillment of the condition that the difference between the input flow (Qi) and output flow (Qs) is equal to or less than 2.5% of Qi. Once this condition is met, the model varies the Qi, taking the flow defined in the next step. The downstream or outlet conditions were defined based on field data as supercritical flow, since this is the normal condition of the stream under study.

For the determination of the roughness coefficient (n) in the section, a calibration of the model was performed using as a comparison parameter, the water level reached for a gauged flow in a specific and referenced cross-section [14]. The process consisted of generating a first run of the hydraulic model with a random "n" and comparing the simulated water-level result (NAS) against the observed water level (NAO), entering a calibration cycle, so that if NAS > NAO, the "n" value decreased and if NAS < NAO, the "n" value increased.

From the calibrated hydraulic model, the water regime of the base period and the two CCS were run. The differences in the hydraulic variables of depth, velocity, and width were obtained from the comparison (using raster methods) of the base scenario and each of the CCS [15].

Figure 3 shows a flow diagram indicating the order of each of the processes carried out.

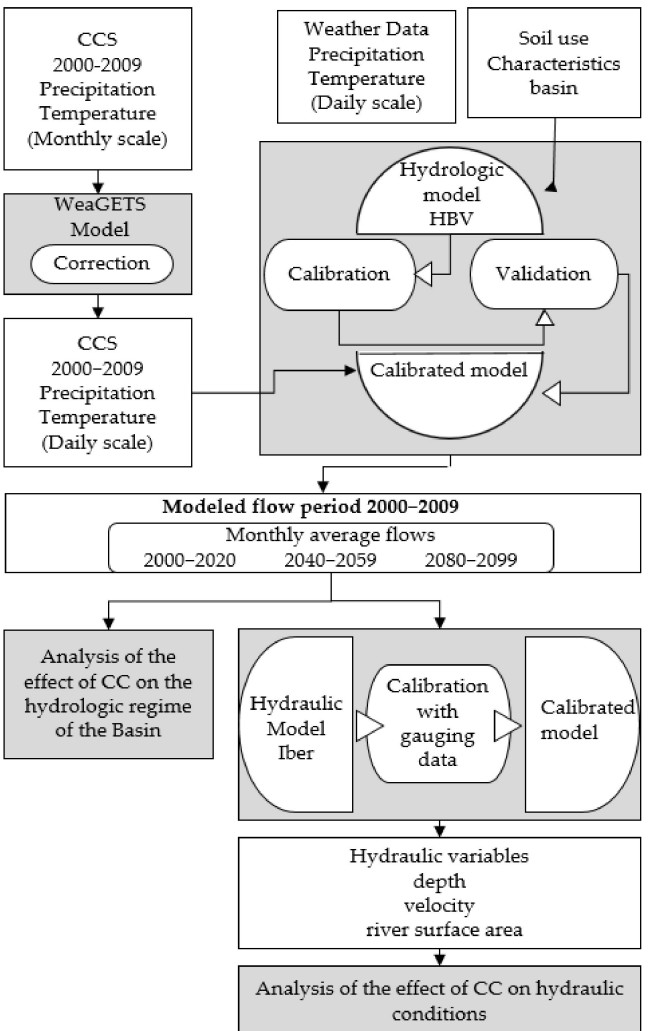

**Figure 3.** Project flow diagram.

### 3. Results

*3.1. Hydrologic Modeling*

The basin, delimited at the gauging point of the Pejibaye River, has a total area of 26,020.00 ha and a perimeter of 99.52 km. Its altitude range is between 614 m.a.s.l. and 2699 m.a.s.l.; the average slope of the main channel is 0.0471 m/m, with an average elevation of 1286 m.a.s.l. The evaluation of the land-use classification indicated a kappa coefficient of 0.695, which featured good reliability. The basin is known for having a forest cover of over 90%, which is not surprising, since there are different types of protected area along the basin.

In the lower areas, the basin is characterized by agricultural land use, mainly sugarcane and pasture (pasturelands), where small populations are also located.

The values obtained from the calibration parameters were consistent, revealing that the maximum soil water storage capacity for the forested areas was above the capacity of the agricultural areas, and that these, in turn, were higher than in the areas without vegetation cover (Table 3). Similarly, the obtained results were consistent with the BETA parameter, a dimensionless variable that defined the relative capacity to contribute to runoff production. The higher the value of BETA we obtained, the greater the runoff capacity of the land use. Therefore, it was expected that the BETA of the forested areas would be lower than in the agricultural areas, and lower than in the areas without vegetation cover.

**Table 3.** HBV model calibration parameters for the basin.

| | Parameters | Description | Value | Source |
|---|---|---|---|---|
| Basin | PERC | Percolation from upper to lower response box | 2.9123 | [16] |
| | UZL | Threshold parameter for extra outflow from upper zone | 76.1697 | [17] |
| | K0 | Additional recession coefficient of upper groundwater store | 0.1968 | [17] |
| | K1 | Recession coefficient of upper groundwater store | 0.0451 | [17] |
| | K2 | Recession coefficient of lower groundwater store | 0.1022 | [17] |
| | MAXBAS | Transformation function parameter | 1.4355 | [16] |
| | Cet | Correction factor for potential evaporation | 0.1097 | |
| | PCALT | Elevation correction factor for precipitation | 0.2152 | [16] |
| | TCALT | Elevation correction factor for temperature | 0.5552 | [16] |
| Forest | FC_1 | Maximum water storage in the unsaturated-zone store | 492.1819 | [17] |
| | LP_1 | Soil moisture value above which actual evaporation reaches potential evaporation | 0.6469 | [17] |
| | BETA_1 | Shape coefficient of recharge function | 1.5149 | [17] |
| Crops and pastures | FC_2 | Maximum water storage in the unsaturated-zone store | 69.1006 | [17] |
| | LP_2 | Soil moisture value above which actual evaporation reaches potential evaporation | 0.9280 | [17] |
| | BETA_2 | Shape coefficient of recharge function | 4.3230 | [17] |
| Construction and bare soil | FC_3 | Maximum water storage in the unsaturated-zone store | 67.1437 | [17] |
| | LP_3 | Soil moisture value above which actual evaporation reaches potential evaporation | 0.9837 | [17] |
| | BETA_3 | Shape coefficient of recharge function | 4.6421 | [17] |

In the hydrologic modeling, one year was defined for the model warm-up, and eight years were defined for the calibration and validation in the last year of the series, as illustrated in Figure 4.

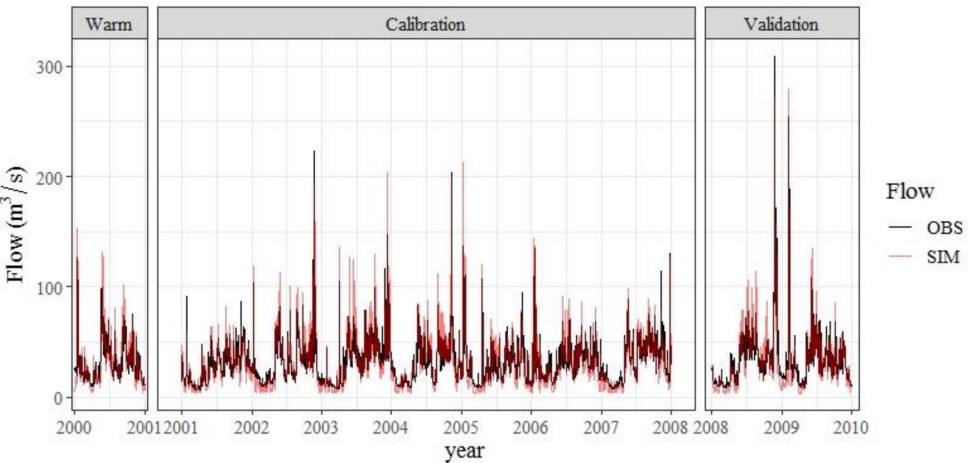

**Figure 4.** Daily flow observed and simulated data for the Pejibaye river basin.

The efficiency of the hydrological model is very good according to the evaluation statistics, $R^2$, *PBIAS*, and *NSE*, used for the time-step scaling. The *NSE* presented values equal to or greater than 0.77 $R^2$ above 0.8 for both the calibration and validation periods, and *PBIAS* of −1.01% and 1.02% for the calibration and validation periods, respectively (Table 4). These *PBIAS* values indicate that the daily flow average was overestimated by 0.32 m$^3$/s for the calibration and underestimated by 0.34 m$^3$/s for the validation.

**Table 4.** Statistical performance of the hydrological model.

| Statistical | Calibration | Validation |
|:---:|:---:|:---:|
| NSE | 0.72 | 0.81 |
| PBIAS | −1.01 | 1.02 |
| $R^2$ | 0.82 | 0.84 |

According to Moriasi et al. [11], the *NSE* and *PBIAS* values obtained rate the model as very good for calibration and validation (Table 2). These model performance ratings allowed us to use them to determine what the response of the basin would be under the corrected CCS (Figure 5).

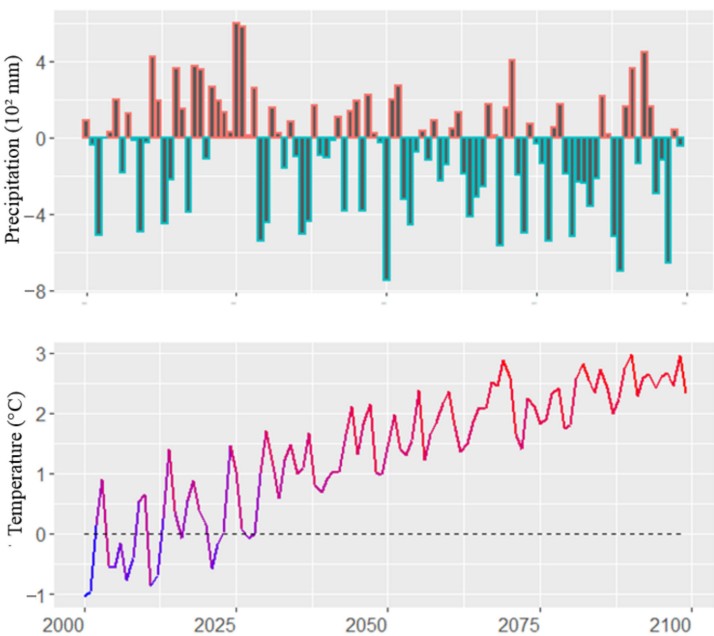

**Figure 5.** Temperature and precipitation anomalies regarding 2000–2019 period of the corrected CSS: Precipitation graph, the green color indicates deficit, and the red color indicates surplus concerning the period 2000-2019. Temperature graph, the blue color indicates temperatures below the average of the reference period and the red color indicates temperatures higher than the reference period.

### 3.2. Effect of Climate Change on the Water Supply

Implementing the calibration values of the different parameters (Table 4) in the HBV model and using the basin's climate-change data series, for both temperature and precipitation (Figure 5), for the period from 2000 to 2099, we generated a daily scale flow series.

The baseline period was from 2000 to 2019, for which the mean temperature was 21.9 °C and the average annual cumulative precipitation was 4584.6 mm. For the comparison periods of 2040–2059 and 2080–2099, the basin presented 1.7% and 3.04% decreases in precipitation and 1.54 °C and 2.5 °C increases in temperature, respectively.

In general, under the CCS, the Pejibaye river basin presented a decrease in the annual water supply in comparison to the 2000–2019 period (baseline). In particular, for the period 2040–2059, a decrease of 71.23 MCM/year was estimated, representing 7.46%, and in the last

two decades of the century, it was estimated to reduce by 117.04 MCM/year, representing a reduction in the current water supply of 12.25% (Figure 6).

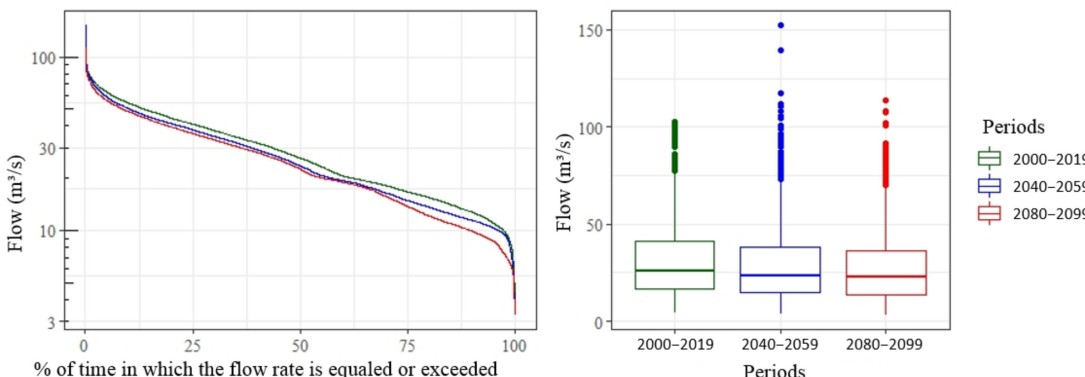

**Figure 6.** Comparison of the flow range of the base period 2000–2019 and the analysis periods 2040–2059 and 2080–2099.

An interannual analysis was conducted to compare the monthly averages, taking as a reference the 2000–2019 values and comparing them with the CCS of the two proposed periods (Figure 7). It was determined that the effects of climate change did not alter the spread of the water supply across different months according to the behavior of the water regime.

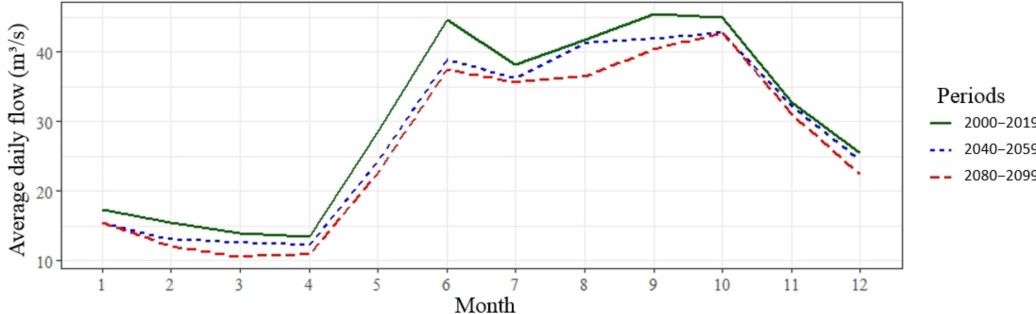

**Figure 7.** The water regime of the basin for the base period and those affected by the CCS.

However, there was a reduction in the annual supply that was spread over the 12 months, although the percentage was much more significant in the first 6 months of the year. Using the water regime data, for the period 2040–2059 the flow reduction was 11.9% in the first half of the year, while for the second half of the year, the reduction was 4.5% (Table 5). We observed similar results for the period 2080–2099, for which the projected reduction was 19.4% for the first 6 months of the year and 8.8% for the second half of the year (Table 5).

### 3.3. Hydraulic Modeling

The chosen channel path had a length of 200 m and was located at the gauging point of the basin (Figure 2). The channel had an average width of 71 m and an average slope of 0.8%. The type of flow was mixed, with a supercritical flow in sections of rapids and a subcritical flow in low pools. The granulometry was very varied, ranging from sands to stones larger than 90 cm. The elevation model generated for the study path is presented in Figure 8.

**Table 5.** Quantification of the effects of climate change on monthly average flows.

| Periods | Jan | Feb | Mar | Apr | May | Jun | Jul | Aug | Sep | Oct | Nov | Dec |
|---|---|---|---|---|---|---|---|---|---|---|---|---|
| Average monthly flow rates for the period 2000–2019 (m³/s) | | | | | | | | | | | | |
| 2000–2019 | 17.35 | 15.47 | 13.92 | 13.40 | 28.61 | 44.73 | 38.27 | 41.80 | 45.49 | 45.06 | 32.86 | 25.57 |
| Reduction in average monthly flow compared to the period 2000–2019 (m³/s) | | | | | | | | | | | | |
| 2040–2059 | −1.92 | −2.37 | −1.32 | −1.20 | −4.32 | −5.91 | −2.01 | −0.55 | −3.59 | −2.27 | −0.58 | −1.14 |
| 2080–2099 | −1.91 | −3.47 | −3.30 | −2.48 | −6.00 | −7.24 | −2.59 | −5.33 | −5.00 | −2.31 | −1.80 | −3.16 |
| Percentage reduction in average monthly flow compared to the period 2000–2019 (%) | | | | | | | | | | | | |
| 2040–2059 | −11.1 | −15.3 | −9.5 | −9.0 | −15.1 | −13.2 | −5.3 | −1.3 | −7.9 | −5.0 | −1.8 | −4.5 |
| 2080–2099 | −11.0 | −22.4 | −23.7 | −18.5 | −21.0 | −16.2 | −6.8 | −12.7 | −11.0 | −5.1 | −5.5 | −12.4 |

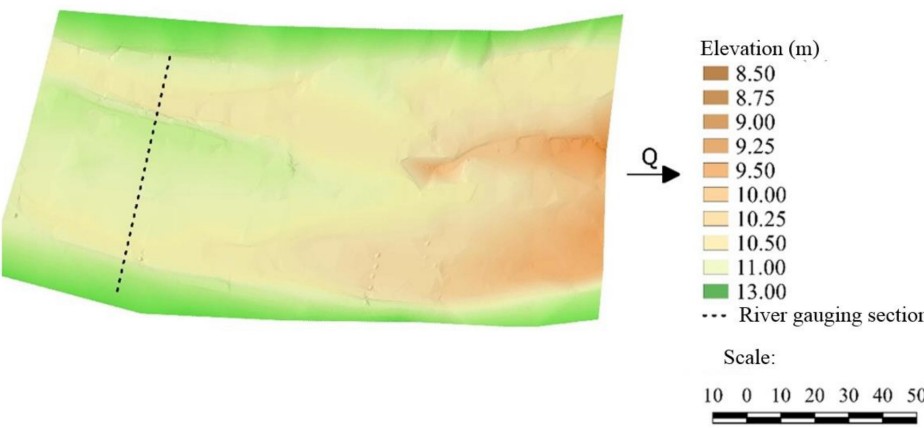

**Figure 8.** Elevation model of the channel path under study.

With the gauging information, the two-dimensional hydraulic model was calibrated by adjusting the Manning's roughness coefficient (n). It was concluded that a value of n equal to $0.063 \text{ s/m}^{1/3}$ generated a simulated depth that was consistent with that observed for a gauged flow of 20.8 m³/s (Figure 9).

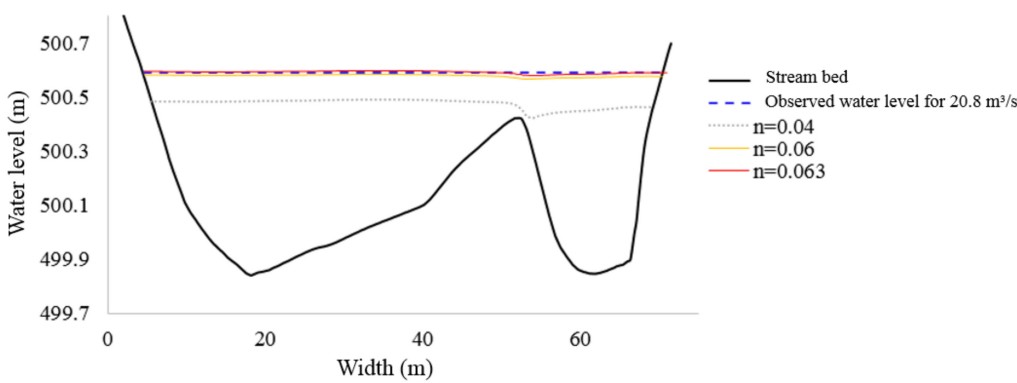

**Figure 9.** Gauging and calibration section for the Pejibaye river channel, calculated and observed water levels used for model calibration as a function of channel roughness.

### 3.4. CCS Effects on Hydraulic Parameters

Variations in flow velocity are related to changes in flow depth. The greatest estimated changes expect for the 2040–2059 period will occur in May, when the mean flow velocity will decrease by 10.9% and the maximum velocities will fall by 36.9% compared to the baseline. For the 2080–2099 period, March, April, and May are estimated to be the months with the greatest variations, with a decrease in mean flow velocity of 11.3% (Figure 10).

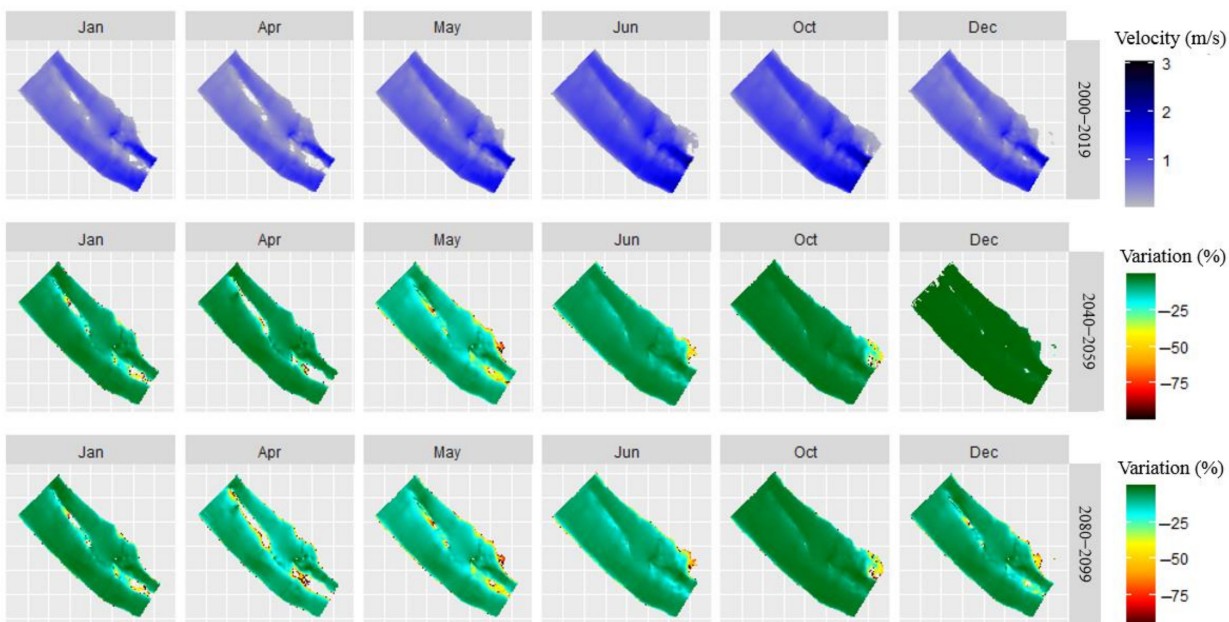

**Figure 10.** Variation in the average monthly flow velocity for the CCS, concerning the period 2000–2019.

For the period between 2040 and 2059, the month presenting the greatest variations compared to the 2000–2019 baseline was May, with an overall decrease in water level of 0.1 m, generating a loss of 1.71 m$^2$ per linear meter of the channel. For 2080–2099, the months of March and June presented the greatest water-level declines, especially April, when the loss of surface width was 2.25 m$^2$ per linear meter of the channel. The greatest impact was expected to occur on the banks and the islets formed in the center of the channel (Figure 11, Table 6).

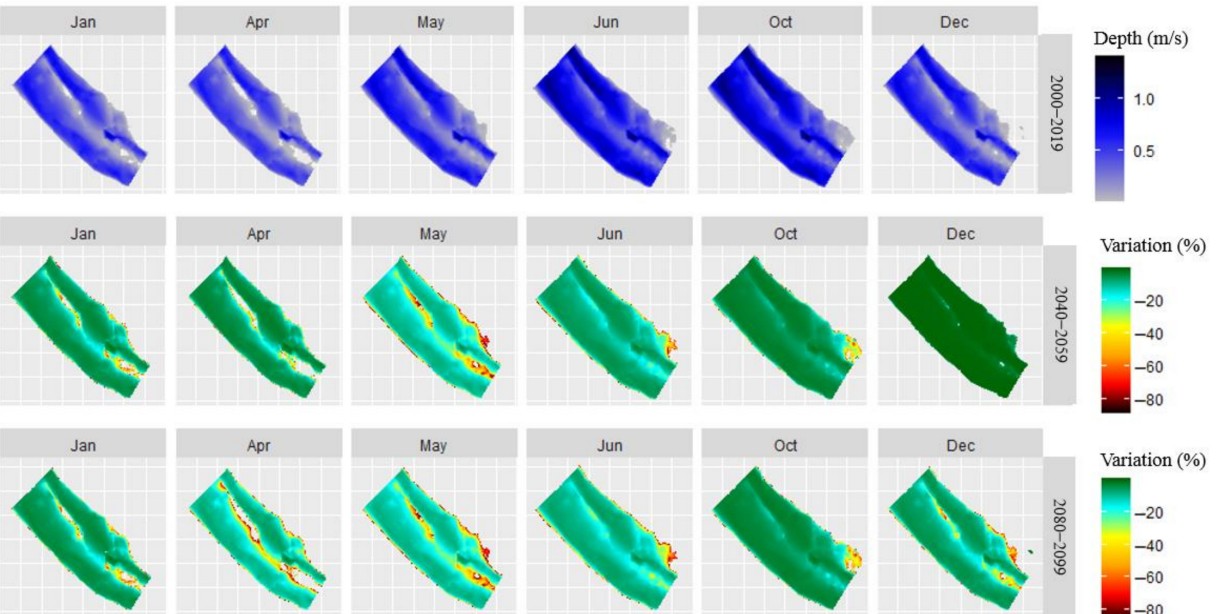

**Figure 11.** Variation in the monthly average depth of flow for the CCS, concerning the period 2000–2019.

**Table 6.** Reduction in the surface width.

| Period | Surface Width Area per Linear Meter of the Watercourse (m$^2$/m) | | | | | |
|---|---|---|---|---|---|---|
| | **Jan** | **Apr** | **May** | **Jun** | **Oct** | **Dec** |
| 2000–2019 | 53.36 | 50.87 | 56.38 | 60.71 | 61.73 | 56.10 |
| | Surface width reduction (m$^2$/m) | | | | | |
| 2040–2059 | −1.08 | −1.08 | −1.71 | −1.57 | −0.62 | −0.03 |
| 2080–2099 | −1.14 | −2.25 | −1.75 | −1.95 | −0.78 | −1.03 |

In both scenarios, the first half of the year shows the greatest variations in velocity, depth, and reduction in the area of the river's surface width. It is projected that, on average, in the first half of the year, the surface width area reduction will be 1360 m$^2$/km for the 2040–2059 period and 1860 m$^2$/km for the 2080–2099 period (Table 6).

## 4. Discussion

Even though only 10 years of data were used to obtain a calibrated hydrological model for the Pejibaye river basin, the evaluation coefficients were good enough (*NSE* = 0.81, *PBIAS* = 1.02, $R^2$ = 0.84) to indicate that the tested tool makes it possible to simulate the hydrological response resulting from the suggested climate-change scenario [18].

In the first mid-century scenario (2040–2059), the temperature will increase by 1.53 °C, and the precipitation will decrease by 1.7% compared to the averages of the reference period (2000–2019). The second scenario, established at the end of the century (2080–2099), will increase these factors by 2.5 °C and reduce precipitation by 3.04% compared to the reference period. These temperature and precipitation variations are consistent with the projections of the Regional Water Resources Committee [3].

It was possible to demonstrate that the increase in temperature conditions and the decrease in precipitation will be reflected in the flow regimes on the basin, which will decrease, particularly during the first half of the year, with the supply reducing by 12.2% for the 2040–2059 period and by 18.8% for the 2080–2099 period compared to 2000–2019 period. This decrease in supply due to the effects of increased temperature and decreased precipitation is consistent with other studies conducted in Central America [19,20]. Zuleta et al. [8] conducted a study in the same basin for a shorter period, but used a scenario in which the average temperature increased by 2.25 °C, estimating an average flow of 24.3 m$^3$/s (25% reduction), which was similar to the value estimated in our study, and of 26.02 m$^3$/s for the second period.

Finally, it was found that, under moderate climate-change scenarios, even a basin with a large protected area (90%) will experience a decrease in its water supply (−8.25% for 2040–2059, −13.86% for 2080–2099). These CC effects result in a decrease in the average depth and velocity of the river flow and, consequently, the stream-width surface will be reduced. In the present study, it was found that for the mid-century (2040–2059), May, and for the end of the century (2080–2099), April will have the greatest decreases in river surface area, of 1710 m$^2$/km and 2250 m$^2$/km, respectively.

**Author Contributions:** Conceptualization, F.W.-H., I.G.-A., L.C.-P. and F.Q.-A.; methodology, F.W.-H., I.G.-A., L.C.-P. and F.Q.-A.; software, F.W.-H.; validation, F.W.-H. and I.G.-A.; formal analysis, F.W.-H., I.G.-A., L.C.-P. and F.Q.-A.; investigation, F.W.-H., I.G.-A., L.C.-P. and F.Q.-A.; resources, F.W.-H., I.G.-A., L.C.-P. and F.Q.-A.; data curation, F.W.-H.; writing—original draft preparation, F.W.-H. and I.G.-A.; writing—review and editing, F.W.-H.; visualization, F.W.-H.; supervision, F.W.-H., I.G.-A., L.C.-P. and F.Q.-A.; project administration, F.W.-H.; funding acquisition, F.W.-H., I.G.-A. and L.C.-P. All authors have read and agreed to the published version of the manuscript.

**Funding:** This research received no external funding.

**Acknowledgments:** Dirección de Agua del MINAE; Instituto Costarrice*NSE* de Acueductos y Alcantarillados. Teaching assistants: Ing. Karen Monge, Ing. Nataly Gómez, Ing. Valeria Serrano, Natalia Hidalgo, Ing. Leandro Peñaranda, Grace Valle Rodríguez, Ing. Sergio Guillén, Daniela Monge.

**Conflicts of Interest:** The authors declare that they do not have a conflict of interest with any of the content reported in this manuscript.

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
