# Peer review of "The Effect of Climate Change on the Water Supply and Hydraulic Conditions in the Upper Pejibaye River Basin, Cartago, Costa Rica"

_hydrology, doi:10.3390/hydrology9050076_

Round 1
Reviewer 1 Report
In the article, the authors described the impact of climate change on the condition of the water supply and hydraulics of the river bed. They did their research on the Pejibaye River. The analysed catchment is characterized by high forest cover, elevation height and high rainfall. It is little transformed anthropogenically (a large part of it lies in the national park).
The methodology is selected and performed correctly. The model has been calibrated and validated. The model was calibrated using the statistical measures: NSE, PBias and R2. It is a pity that their formulas were not given - in my opinion, it would be good to write them. Two models were used: HBV (for hydrology) and IBER (for hydrodynamics).
The literature is somewhat scarce, especially concerning numerical modelling of hydrodynamic conditions. Suggestions for some articles, perhaps worth adding:
- Plesiński K., Radecki-Pawlik A., Michalik P. 2017. On using HEC-RAS model for river channel changes predictions along the engineered Czarny Dunajec river. Scientific Review Engineering and Environmental Sciences, 26(3), 346-360, doi.org/10.22630/PNIKS.2017.26.3.34
- ArdıçlıoÄŸlu M., Kuriqi A. 2019. Calibration of channel roughness in intermittent rivers using HEC‑RASmodel: case of Sarimsakli creek, Turkey. SN Applied Sciences, 1:1080, doi.org/10.1007/s42452-019-1141-9
The list of literature should also be standardized.
Other remarks:
- There are many abbreviations given in table 3 - it would be good to explain them.
- Chapter 2.2, lines 94-96 - The authors write: "The model was supplied with daily data on temperature (°C), precipitation (mm) and flow (mm / day), monthly averages of temperature and evapotranspiration, and a matrix of land-use area fraction by elevation zone." Instead of "flow", there should be "precipitation". The unit "mm / day" corresponds to the precipitation. For flow, it should be "m3 / s".
- Figures 10 and 11 should be bigger and clearer.
Generally. Despite these minor mistakes, the work is interesting.
Author Response
Dear Reviewer
We thank you for your comments. Your observations were considered and integrated into the study. Below, we present the answers to your contributions in the same order as you did.
- The model has been calibrated and validated. The model was calibrated using the statistical measures: NSE, PBias and R2.It is a pity that their formulas were not given - in my opinion, it would be good to write them.
The formulas of the statistical measures used: NSE, PBIAS, and R2, were added in the "2.2. Hydrologic Model" section in "Materials and Methods".
- The literature is somewhat scarce, especially concerning numerical modelling of hydrodynamic conditions. Suggestions for some articles, perhaps worth adding:
- Plesiński K., Radecki-Pawlik A., Michalik P. 2017. On using HEC-RAS model for river channel changes predictions along the engineered Czarny Dunajec river. Scientific Review Engineering and Environmental Sciences, 26(3), 346-360, doi.org/10.22630/PNIKS.2017.26.3.34
- ArdıçlıoÄŸlu M., Kuriqi A. 2019. Calibration of channel roughness in intermittent rivers using HEC‑RASmodel: case of Sarimsakli creek, Turkey. SN Applied Sciences, 1:1080, doi.org/10.1007/s42452-019-1141-9
The two suggestion articles were added to the document, in addition to another article (reference 13) to improve the literature on this subject.
- The list of literature should also be standardized.
The list of literature references cited in the paper was reviewed, corrected, and standardized, according to the instructions of the “Reference List and Citations Style Guide” for MDPI Journals.
- There are many abbreviations given in table 3 - it would be good to explain them.
An extra column was created in Table 3 describing each of the abbreviations.
- Chapter 2.2, lines 94-96 - The authors write: "The model was supplied with daily data on temperature (°C), precipitation (mm) andflow (mm / day), monthly averages of temperature and evapotranspiration, and a matrix of land-use area fraction by elevation zone." Instead of "flow", there should be "precipitation". The unit "mm / day" corresponds to the precipitation. For flow, it should be "m3 / s".
Correct, the unit for flow was changed to "m3/s" in the text.
- Figures 10 and 11 should be bigger and clearer.
These figures (10 and 11) were enlarged for a better visualization.

Reviewer 2 Report
Please address these comments and suggestions:
Line 38: fix text reference “2008”
Line 48: fix reference “y” should be “and”
Line 55: change sentence: annual average temperature….. mean precipitation
Line 62: Which Either basin are the authors referring to?
Line 73: River 1? Is this a typo?
Line 80: Figure 1 not 2
There is room for improvement in the introduction.
Figure 1 caption can be improved. Also Map legend please change ELev to Elevation (m), use English only “Proyection”
Table1: What is the parameter distribution in the MC method? Uniform? Triangular? Include information in table.
Line 138: Afterwar? Please revise. ECC? What is this not mentioned anywhere in the text. And the WeaGets? Is it not MulGets? According to the reference provided?
Line 139 this paragraph is very confusing. Input information to what?
Line 154: hydrological model not models.
Line 178: Its Figure 3 not 4. Also, in figure 3 what is gauning data? Is it gauging?
Line 212: R2 not r2. Also, the PBIAS is positive shouldn’t the model be underestimating the flow? Instead of overestimating? As referred previously in the method section?
Please revise figure 5 caption.
Line 240-241: what is P3? Another acronym that is not referenced anywhere?
Line 247: what is MM? millimetres? mm instead
Figure 6 caption: base period 2000-2019 not periods.
Line 287 0.063 s/m 1/3? What does this means?
Line 318: Table 6 I presume?
Discussion: If the calibration is underestimating the flow in approximately 4% how will this influence the interpretation of the results in the future when mentioning a decrease in water supply of -8.25%? How great of impact is it really?
Conclusions: Maybe merge the discussion and conclusions or redo the entire conclusions section since the first two paragraphs are already written in the discussion.
References:
REF 6: change capital letters
REF 8: change capital letters
Ref 10 should be like this:
Moriasi, D.N., et al. (2007) Model Evaluation Guidelines for Systematic Quantification of Accuracy in Watershed Simulations. Transactions of the ASABE, 50, 885-900.
http://dx.doi.org/10.13031/2013.23153
Author Response
Dear Reviewer
We thank you for your comments. Your observations were considered and integrated into the study. Below, we present the answers to your contributions in the same order as you did.
- Line 38: fix text reference “2008”.
The reference [2] was corrected with the correct year of publication, corresponding to 2022.
- Line 48: fix reference “y” should be “and”.
The reference [4] that had "y" for "and" was corrected.
- Line 55: change sentence: annual average temperature…. mean precipitation
The sentence was reworded to clarify the idea and corrected the "annual average..." to "annual mean temperature..."
- Line 62: Which Either basin are the authors referring to?
The sentence was restructured to clarify the idea of the paragraph. The authors meant "Either basin" to refer to basins located in regions with less anthropogenic impact, such as the basin under study in this paper.
- Line 73: River 1? Is this a typo?
Yes, it is a typo. The line was corrected.
- Line 80: Figure 1 not 2
Corrected for Figure 1.
- There is room for improvement in the introduction.
Some of the paragraphs of the introduction were rewritten to improve the text comprehension.
- Figure 1 caption can be improved. Also Map legend please change ELev to Elevation (m), use English only “Proyection”
A new caption was written for Figure 1. Also, the legend of "ELev" was changed to "Elevation" and the map projection was changed to WGS84 EPSG:4326.
- Table1: What is the parameter distribution in the MC method? Uniform? Triangular? Include information in table.
The parameter distribution in the Monte Carlo method was added in section "2.2. Hydrologic Model", this method uses a uniform distribution within the ranges given for each parameter.
- Line 138: Afterwar? Please revise. ECC? What is this not mentioned anywhere in the text.
The "Afterwar" and "ECC" were a typo. They were corrected to "Afterwards" and "CCS" for climate change scenarios, respectively.
- And the WeaGets? Is it not MulGets? According to the reference provided?
The reference provided on the subject of WeaGets was changed and another article was cited to avoid confusion about the tool used in this project.
- Line 139 this paragraph is very confusing. Input information to what?
Once the corrections to questions 10 and 11 were addressed, the paragraph was also clarified, improving the interpretation of the text. As for the "input information" this is the P and T data observed in the period 2000-2019; in the text this was addressed and reworded to specify which type of input information was being referred to.
- Line 154: hydrological model not models.
"Hydrological models" was corrected to "Hydrological model", as well as in other lines of the article where it was written incorrectly.
- Line 178: Its Figure 3 not 4. Also, in figure 3 what is gauning data? Is it gauging?
Corrected by Figure 3. Also, the word "gauning" was corrected to "gauging" in this same figure.
- Line 212: R2 not r2. Also, the PBIAS is positive shouldn’t the model be underestimating the flow? Instead of overestimating? As referred previously in the method section?
All “r2” in the article were corrected to “R2”.
Correct, the values of PBIAS, as well as R2 and NSE were edited, since those in the text were erroneous. The new NSE presents values greater than 0.77, R² above 0.8 for both the calibration and validation periods, and the PBIAS of -1.01% and 1.02% for the calibration and validation periods respectively (see Table 4). This indicates that the efficiency of the hydrological model is very good, in addition to supporting what is described in the "Materials and Methods" section about these three statistical evaluations.
- Please revise figure 5 caption.
The caption in Figure 5 was modified.
- Line 240-241: what is P3? Another acronym that is not referenced anywhere?
"P3" is a typo, this was removed from the article.
- Line 247: what is MM? millimetres? mm instead
"MM3" was to indicate "million cubic meters", however, to avoid any misunderstanding among readers, this unit was changed in the paper to "MCM" which in civil engineering is the abbreviation for 1 000 000 m3.
- Figure 6 caption: base period 2000-2019 not periods.
The caption in Figure 6 was corrected to "base period 2000-2019".
- Line 287 0.063 s/m 1/3? What does this means?
The value of 0.063 is Manning's roughness coefficient of the hydraulic model, whose units are s/m1/3, this unit was misspelled in the document (“s/m 1/3”) and was corrected as "s/m1/3".
- Line 318: Table 6 I presume?
Correct, it referred to Table 6, this was fixed in the article.
- Discussion: If the calibration is underestimating the flow in approximately 4% how will this influence the interpretation of the results in the future when mentioning a decrease in water supply of -8.25%? How great of impact is it really?
The algorithm used to calculate the PBIAS was analyzed and corrected (as described in the answer to question 15), obtaining a calibration value of -1.01% (not 4% as was erroneously stated in the text) and validation of 1.02%. Since we are working with medium and long-term Climate Change Scenarios, we considered not quantifying the uncertainty or the risk in the amounts of water supply decrease.
- Conclusions: Maybe merge the discussion and conclusions or redo the entire conclusions section since the first two paragraphs are already written in the discussion.
The "Discussion" and "Conclusions" were merged into the "Discussion" section to avoid repetition in the text. The “Conclusions” section was removed from the article, as it is not mandatory according to the journal's instructions.
- References: REF 6: change capital letters, REF 8: change capital letters, Ref 10 should be like this: Moriasi, D.N., et al. (2007) Model Evaluation Guidelines for Systematic Quantification of Accuracy in Watershed Simulations. Transactions of the ASABE, 50, 885-900. http://dx.doi.org/10.13031/2013.23153
The list of references (including ref: 6, 8, and 10) has been revised and corrected according to the instructions of the “Reference List and Citations Style Guide” for MDPI Journals.

Round 2
Reviewer 2 Report
In Figure 1 Please change "Proyection" with "Projection"
Author Response
Dear Reviewer
We are very grateful for the recommendations. The correction of the word projection was made in Figure 1.
Kind regards